# Natural Compounds as Elicitors of Plant Resistance Against Diseases and New Biocontrol Strategies

**Agnieszka Jamiołkowska** 

Department of Plant Protection, Faculty of Horticulture and Landscape Architecture, University of Life Sciences in Lublin, 20-069 Lublin, Poland; aguto@wp.pl; Tel.: +48-8-1524-8132

**Abstract:** The goal of sustainable and organic agriculture is to optimize the health and productivity of interdependent communities of soil life, plants, animals, and people. Organic plant production uses natural products and natural self-regulation processes occurring in the ecosystem. The availability of innovative applications and molecular techniques opens up new possibilities in the approach to plant protection for sustainable and organic agriculture. New strategies not only directly protect plants against pathogens but can also induce enhanced immunity that permanently protects against pathogenic strains. This review focuses on the bioactive properties of selected natural compounds (of plant and animal origin), their action on pathogens, and their roles in the mechanism of inducing plant resistance. The author presents selected activities of organic bioactive compounds, such as allicin, naringin, terpenes, laminarin, carrageenans, chitin and chitosan, and outlines the possibilities for their application in protecting crop plants against diseases. In addition, this mini review describes the mechanism of action of the above compounds as elicitors of defense reactions in the plant and the possibility of their utilization in the production of biological preparations as elements of a new plant protection strategy.

**Keywords:** crop production; allicin; chitosan; laminarin; naringin; terpenes; elicitors; plant resistance

## 1. Introduction

Crop plants are constantly attacked by pathogens during both pre- and post-harvest stages, often causing economically important yield losses. Sustainable and organic farming practices require proper environmentally friendly disease management to improve the quality and quantity of agricultural outputs. The indiscriminate utilization of pesticides in crop protection leads to a number of disadvantages to both target and non-target organisms, as well as to environmental pollution. Due to the residual problem and toxicity to the living environment, chemical pesticides are not suitable for crop production. Therefore, products of plant and animal origin have recently gained enormous importance in the quest to develop better alternatives to chemical pesticides by considering multiple modes of action against plant diseases [1–5].

Modern plant protection is based on the sustainable use of pesticides, mainly the application of non-chemical methods of plant protection against diseases, pests and weeds. The organic production system uses biological and physiological plant mechanisms supported by the rational use of conventional, natural, and biological preparations [6–10]. In view of new problems emerging in the field of plant protection, there is an urgent need to develop more effective, sustainable and environmentally friendly tools for pathogen control.

## 2. Natural Bioactive Compounds as Elicitors

Natural bioactive compounds are substances of plant and animal origin. They modulate plant growth and are involved in plant defense responses, including limiting pathogen development.

These compounds usually belong to one of three large chemical classes: terpenoids, phenolics, and alkaloids [11]. The action of these natural compounds is not specific, and their effect on pathogens is versatile. Natural bioactive compounds used in plant protection kill pathogens (fungicidal effect) or limit their development (fungistatic effect), as well as induce plant defense reactions as elicitors [12]. Each molecule/compound that triggers or stimulates certain defense mechanism in a plant is called an elicitor. As a result of the interaction of an elicitor with a receptor of the cell on which it acts, a metabolic stimulus, called a "signal", is created, due to the possibility of its movement intracellularly, as well as intercellularly and systemically. Plants sprayed with these compounds react quickly: membrane receptors of plant cells bind elicitor molecules, induce local resistance, and subsequently generate plant molecular response [13]. Elicitors are classified as physical or chemical, biotic or abiotic, and complex or defined depending on their origin and molecular structure [14]. Elicitors derived from extraneous microbes are called microbe-associated molecular pattern (MAMP)-type molecules (microbe-associated molecular pattern), and those derived from pathogenic organisms, PAMP-type molecules (pathogen-associated molecular pattern). Activation of basal immunity may also involve molecules derived from the cells of the attacked plant, released following a phytopathogen attack, or produced under stress, called DAMP-type molecules (damage/danger-associated molecular pattern) [15–17]. The perception of MAMP/PAMP and DAMP molecules is mediated by receptor proteins located in the plasma membrane generally referred to as PRRs (pattern recognition receptors) [18,19], which recognize a particular molecular pattern (signature) of a foreign or own molecule. In agricultural practice, elicitor treatments of plants in the absence of virulent pathogens cause MAMP/PAMP or DAMP molecules, bound to the membrane receptor, to activate intracellular signaling pathways that trigger a broad stream of defense responses in the plant, called priming or PAMP-triggered immunity (PTI)-type local immunity [20]. Thus, in plant defense strategies, immunity activated by MAMP/PAMP and DAMP molecules is the first line of local defense, thanks to which the plant can not only counteract infections, but also actively fight emerging pathogenic organisms, often resistant to chemical pesticides. Therefore, priming is defined as the physiological status of plants leading to faster and stronger activation of defense responses to subsequent biotic and abiotic stresses [21–23]. This defense reaction aims to restrict intruder growth and can lead to induced systemic resistance (ISR) or systemic acquired resistance (SAR), making the plant less susceptible to subsequent pathogen attack [17]. Triggering of a permanent resistance mechanism in plant ontogeny development is one of the methods of plant protection, especially due to its pro-ecological nature.

## 3. Natural Compounds Against Plant Diseases

To date, many bioactive compounds have been isolated, and some of them have contributed to the development of novel plant-based biopesticides for food production. The appropriate selection of biomolecules for the creation of biopesticides with multiple modes of action against target pathogens is a safer alternative for sustainable and organic production.

Allicin (diallyl thiosulfinate) is an organosulfur compound obtained from garlic with antibacterial and antifungal activity [24]. The antifungal properties of allicin against various pathogenic fungi have been described by many authors (Table 1) [6,25,26]. Garlic juice used *in vitro* limits the growth of bacteria (genus *Agrobacterium*, *Erwinia*, *Pseudomonas*, *Xanthomonas*) and fungi (*Alternaria alternata, Fusarium moniliforme, Cercospora arachidicola, Colletotrichum coccodes, Botrytis cinerea, Rhizoctonia solani*) [6,26,27]. Many authors have reported high efficiency of allicin under field conditions [28–30]. Spraying sweet pepper plants several times with a solution of garlic pulp extract (Bioczos Liquid preparation) improved plants' health and was more effective than azoxystrobin treatment (Amistar 250 SC fungicide) [26]. Naringin (4′,5,7-trihydroxyflavanone-7-β-d-α-l-rhamnosyl(1→2)-β-d-glucoside) is an organic compound of plant origin [31]. It is one of the most common citrus flavanone glycosides, which is mainly found in the grapefruit pulp and seeds, as well as in the epidermis of lemon and orange (*Citrus* L.). Naringin antimicrobial activity is related to the presence of many biologically active compounds contained in the pulp and seeds of grapefruit (Table 1). Compounds present in

the grapefruit extract are not only endogenous flavonoids and glycosides (mainly naringin), but also terpenes, coumarins and furanocoumarins [31]. Many authors have demonstrated the effectiveness of grapefruit extract in the field against gray mold, fusariosis, and alternariosis in vegetable and ornamental plants (*Phytophthora cryptogea*, *P. cinnamomi*, *Fusarium oxysporum* f. sp. *cyclaminis*), as well as in soybean crops (*Phomopsis phaseoli*, *Fusarium* spp., *Sclarotinia sclerotiorum*, *Phoma exigua*) [3,32,33]. Grapefruit extract (preparation Biosept 33 SL) limited the development of alternariosis on potato grown in the ecological system. Several treatments also limited the development of *Cercospora beticola* on beetroot in the organic plantation [34]. Terpenes (mainly terpinen-4-ol, gamma-terpinene, 1,8-cineole) are organic chemical compounds contained in tea tree oil (*Malaleuca alternifolia* L.). The oil is obtained from leaves and small branches of the tree, which grows in Australia. Tea tree oil has a strong antiseptic effect and is used in the control of phytopathogenic fungi and bacteria by destroying their membranes and cell organelles (Table 1) [35–37]. Laboratory and field studies show the high efficiency of tea tree oil (preparation Timorex Gold 24 EC) in limiting *Bremia lactucae* on lettuce and high effectiveness in protecting this plant against downy mildew [38]. Many biologically active compounds, acting as defense elicitors, have been found in marine algae extracts [39,40]. Marine algae are a rich source of organic compounds such as amino acids, vitamins, enzymes, mono-, oligo- and polysaccharides ($\beta$-1,3-glucan), phytohormones (including auxins and gibberellins), as well as organic compounds of boron, iron, zinc, molybdenum, manganese and iodine [9,39,41]. Many authors pointed to the antibacterial, antifungal and antiviral properties of marine algae extract (Table 1) [1,41]. Chitin is a polysaccharide present in the natural state as an ingredient in the shells of marine crustaceans, and the main component of cell walls of filamentous fungi. This organic compound is obtained by chitosan distillation using sodium hydroxide at elevated temperature or by means of enzymatic reactions [42]. Research has shown the activity of this organic substance against viruses, bacteria, fungi and other pathogens (Table 1) [43]. Field studies confirmed the effectiveness of chitosan micro-gel (0.1%) in inhibiting of soybean diseases caused by soil-borne pathogenic fungi (*Fusarium* spp., *Pythium* spp., *Botrytis cinerea*, and *Rhizoctonia solani*) [3]. Chitin is known as a strong fungal microbe-associated molecular pattern (MAMP) molecule, which is recognized by plants and which activates their immune response [44–46].

**Table 1.** List of natural bioactive compounds (origin and form) and their activity against pathogens. MAMPs; microbe-associated molecular patterns.

| Origin | Species Name | Form | Bioactive Natural Compound | Effective Against | Effects Shown | References |
|---|---|---|---|---|---|---|
| Vascular Plants | *Allium* spp. | garlic pulp | allicin | *Agrobacterium tumefaciens, Pseudomonas syringae* pv. *maculicola, P. syringae* pv. *phaseolicola, Ervinia carotovora, Escherichia coli* | *in vitro* | [24,25] |
| | | | | *Alternaria alternata, Botrytis cinerea, Colletotrichum coccodes, Rhizoctonia solani* | sweet pepper, strawberry | [6,26,27] |
| | | | | *Pythium aphanidermatum, Phytophthora infestans* | *in vitro*, potato tubers | [24,25,47] |
| | *Citrus* spp. | grapefruit extract | naringin | Gram-positive pathogenic bacteria | *in vitro* | [31] |
| | | | | *Cercospora arachidicola* | onion | [7] |
| | | | | *Pythium oligandrum*, soil-borne fungi | bean, pea soybean | [48] |
| | | | | *Fusarium culmorum, F. oxysporum, F. solani, Rhizoctonia solani, Sclerotinia sclerotiorum* | common bean, pea | [2] |
| | | | | *Fusarium semitectum, Aspergillus flavus, Aspergillus parasiticus, Penicillium expansum* | *in vitro* | [49] |
| | *Malaleuca alternifolia* L. | tea tree oil | terpenes | *Xanthomonas vesicatoria* | *in vitro* | [4] |
| | | | terpenes (α-terpineol, terpinolene, 1,8-cineole) | *Botrytis cinerea, Aspergillus fumigatus, Chaetomium globosum, Penicillium chrysogenum* | *in vitro* | [35,37] |
| | | | terpinen-4-ol, γ-terpinen, 1,8-cineole | *Blumeria graminis* f. sp. *hordei, Fusarium graminearum, F. culmorum, Pyrenophora graminea* | *in vitro* | [36] |
| | | | terpenes | *Ascochyta rabiei, Colletotrichum lindemuthianum, Drechslera avenae, Alternaria radicina, A. dauci* | *in vitro* | [50] |
| Protists | brown algae (*Ascophyllum nodosum*) | | β-1,3-glucan laminarin | *Pyrenophora teres, Rhynchosporium secalis, Puccinia hordei* | spring barley, oat | [51] |
| | brown algae (*Laminaria digitata*) | | | *Botrytis cinerea, Plasmopara viticola* | grapevine | [13] |
| | several species of marine algae | algae extract | | *Staphylococcus aureus, Escherichia coli, Bacillus subtilis, Streptococcus aureus, Proteus subtilis* | several plants | [1] |
| | | | | pathogenic fungi, *Rhizoctonia solani, Botrytis cinerea, Phytophthora cinnamomi* | several strawberry | [1,39] |
| | marine algae (*Spatoglossum variabile, Melanothamnus afaqhusainii, Halimeda tuna*) | | | *Macrophomina phaseolina, Fusarium solani, Rhizoctonia solani, Verticillium* spp. | sunflower | [9] |
| | red algae (*Chondrus crispus, Gigartina stellata*) | | carrageenans | Tobacco mosaic virus (TMV), *Pectobacterium carotovorum, Botrytis cinerea* | tobacco | [52,53] |
| | seaweeds | | | *Xanthomonas oryzae* pv. *oryzae* | *in vitro* | [54] |
| Animals | marine crustaceans (*Crustacea*) | | chitosan, chitin (MAMPs) | *Botrytis cinerea* | strawberry | [55] |
| | | | | *Alternaria alternata, Botrytis cinerea, Fusarium* spp., *Pythium* spp., *Rhizoctonia solani* | peppermint, soybean | [3,56] |
| | | | | *Sclerotinia sclerotiorum, Cladosporium cladosporioides* | lemon balm, peppermint | [56] |
| | | | | pathogenic bacteria and fungi | several plants | [43] |

*Mode of Action of Natural Elicitors*

Many natural compounds act as elicitors of defense responses in plants [57–59]. Organic elicitors act by inducing systemic acquired resistance (SAR) [30]. Indirect action of bioactive compounds (as extract/oil) in plant cells stimulates the release of protein and lipid elicitors. Synthesis of phytoalexins and pathogenesis-related (PR) proteins, accumulation of callose and cell wall lignification, as well as enhanced activity of various defense enzymes is initiated in plant cells, which protects plants against pathogens (Figure 1) [58].

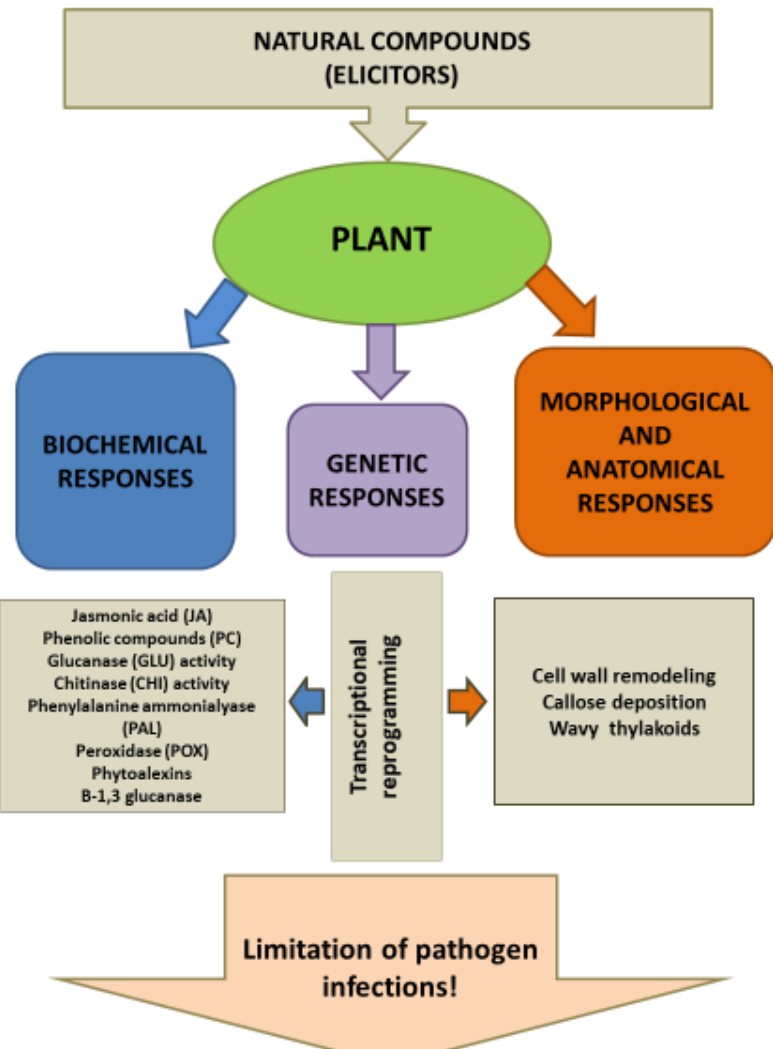

**Figure 1.** Immune response of plant under the influence of natural elicitors [58].

Phenolic compounds belong to the group of phytoalexins synthesized under the influence of elicitors. Several preventive sprayings of plants with grapefruit and algae (*Ascophylum nodosum*) extract increased the content of phenolic compounds in the aerial part of *Capsicum annuum* L. [33]. Many authors [59–61] have reported that the use of the above-mentioned extract can trigger a number of defense reactions in plants (production and accumulation of phenolic compounds), as a result of plant immunization. Phenolic compounds inhibit sporulation, spore germination and the growth of germinative hyphae of pathogenic fungi [58,62]. Recent studies have indicated that flavonoids produced by the plant are also present in plant root secretions, acting on pathogenic rhizosphere microorganisms [61]. Secondary metabolites synthesized by plants have a protective function against the "attacking" pathogens. The accumulation of phenolic compounds in a plant is the result of

qualitative changes in the respiration process. The biological activity of these compounds is associated with their ease of oxidation [63]. They act as electron carriers in the final oxidation reactions catalyzed by phenol oxidases. Particularly important in reactions against pathogens are not phenols themselves, but the products resulting from their transformation and oxidation, e.g. quinones and chlorogenic acid. The presence of chlorogenic acid was associated with potato resistance to common scab caused by *Streptomyces scabies*, while its content in carrot roots increased more than twice after *Thielaviopsis basicola* infection [63].

Linear β-1,3-glucan laminarin derived from brown algae *Laminaria digitata* elicits a variety of defense reactions in tobacco plants, such as stimulation of phenylalanine ammonia lyase, caffeic acid O-methyltransferase and lipoxygenase, as well as the accumulation of salicylic acid and PR proteins. Furthermore, certain glucans have been reported to enhance resistance against viruses and bacteria [13,64]. The SAR mechanism has been observed after applying algae extract (*Lichnis viscaria* L.) to plants [31]. Described defense responses included activation of mitogen-activated protein kinases, $Ca^{2+}$ influx, oxidative burst and alkalization of the extracellular medium. Applied to tobacco or grapevine plants, laminarin induced the accumulation of phytoalexins and expression of a set of PR proteins [13,64].

Molecular mechanisms underlying chitin action have been described in *Arabidopsis* sp. and rice. AtCERK1, a trans-membrane protein, is known in *Arabidopsis* sp. as the central component of the chitin receptor in combination with AtLYK5 and/or AtLYK4 kinases. OsCERK1 in *Oryza sativa* does not bind chitin, but interacts with OsCEBiP, which recognizes chitin, a related protein with three LysM domains, but lacking the protein kinase domain. In both *Arabidopsis* and rice, interaction with the chitin octamer appears to lead to receptor dimerization and activation, which in turn triggers immune responses conferring disease resistance [44–46]. Chitosan acts as an elicitor stimulating the production of two phytoalexins: formononetin and calyocasin in *Astragalus membranaceus* hairy roots. Chitosan can cause an oxidative burst that induces the expression of the MPK3 and MPK6 genes related to mitogen-activated protein kinase signaling (MAPK) cascades (Figure 1) [46].

Grapefruit extract, used as a spray on plants, acted as an elicitor of immunity, inducing systemic acquired resistance (SAR) through 7-geranoxycoumarin present in the extract [65].

In addition to numerous biochemical mechanisms, host plant predispositions to form physical barriers play an important role in resistance [63]. Impregnation of cell walls with various substances such as waxes (in epidermis), lignin (in woody cells), and suberin (in cork) forms barriers hindering the penetration. Other types of passive resistance include barriers impeding the movement of a foreign agent (biotic or abiotic), for example, high cellulose content or strong development of sclerenchyma (Figure 1).

If the pathogen is able to overcome constitutive barriers, its spread may be limited by the induced modification and tightening of the cell walls [63]. As a result of plant infection, low-molecular fragments of the cell wall are released. These signals, called DAMPs, mainly include cell wall or extracellular protein fragments, peptides, nucleotides, amino acids, and are detected by plasma membrane-localized receptors of surrounding cells to regulate immune responses against the invading organisms and stimulate damage repair. The role of oligosaccharides as DAMPs in plant defense reactions against pathogens and physical agents have been well covered by several recent reviews [66,67]. Molecule fragments released from cell walls trigger genetic alterations and induce biological activity of plants. Different types of genetic transcription within cutin cells lead to cutin morphological modulation such as higher accumulation of cutin monomer [68], deposition of cell wall polysaccharides [66], callose accumulation, papillae formation, lignification and suberin synthesis, as well as accumulation of structural proteins [67]. Under the influence of elicitors, structural changes occur in association with stiffening of this structure, which limits and blocks pathogen development (Figure 1). Lignification can be a direct cause of pathogens reduction and an important element of hypersensitivity reactions, for example, in potato tubers infected with *P. infestans* [69]. Phenolic acids and phytoalexins are the precursors of lignin synthesis in plant cells attacked by pathogen. Lignin is a polymer that is of

great importance in the body's defense response to pathogen infection. It is resistant to enzymatic degradation by the pathogen and has a different biochemical nature depending on the host plant and type of stress factor [70–72]. Some natural compounds can modify plant cell wall, which in the light of the latest research reacts to external stimuli [33]. Remodeling of the cell wall, such as changes in the thickness and structure of the epidermis, degradation of chloroplasts, and the corrugated system of thylakoids, was noted in plants where defense responses were induced [58]. Marine algae and garlic extract were shown to have an impact on sweet pepper leaf blade growth and thickness [33]. Recent research has demonstrated that the action of natural compounds (elicitors) also leads to a diverse expression of plant genes responsible for triggering defense responses (Figure 1) [23,58].

## 4. Commercial Uses of Natural Elicitors in Organic Plant Production

The use of natural compounds for pathogen control is very attractive, and the availability of novel applications and molecular techniques open new avenues for plant protection approaches. Many organic compounds have already been commercialized and are present on the market as biofertilizers or plant growth biostimulants. These, among others, include Biosept 33SL (grapefruit extract), Bio-Algeen S90 Plus, Labimar 10S, Kelpak SL, Lysodin Alga-Fert (marine algae extract), Bioczos BR (garlic extract), Timorex Gold 24 EC (tea tree extract), and Vaxiplant SL (laminarin) [30,33,36]. Generally, these preparations are biodegradable, non-toxic, non-polluting, and non-hazardous to various organisms. In addition, many of them not only mitigate stress-induced limitations and regulate/modify physiological processes in plants to stimulate growth and increase productivity, but also directly limit the development of phytopathogens [30,73].

The production of preparations based on biological components is difficult, requires a large amount of starting material, and the obtained product is of different quality (variable content of biologically active compounds) and may prove unstable in the production process. This means that organic compounds present in the starting material at the beginning of the manufacturing process are not preserved in the final material during the production process of the preparation. An example of this is the multitude of commercial seaweed extracts, often derived from the same species, that are rarely equivalent [74]. Commercial biostimulants manufactured from similar sources are usually marketed as equivalent products, but may differ considerably in composition, and thus in efficiency [75–77].

Currently, interest in bioactive natural compounds with phytosanitary activity results from their molecular structures that can be modified and new stable molecules can be created. The key to solving the abovementioned problem is the development of new bioproducts through chemical modifications of natural biocompounds. This will contribute to the formation of pure and stable compounds and creation of new biopesticides for widespread use in agriculture [78]. Another advantage of these natural bioproducts is their safety during use and lack of residues. Some of them can be used in mixtures with pesticides, which does not affect their effectiveness [9], while improving efficiency. Due to the pest resistance phenomenon and withdrawal of many active pesticide substances from plant production, preventive use of products based on natural components can and should become an alternative to pesticides.

**Author Contributions:** Conceptualization, software, resources, validation, writing—review and editing, A.J. All authors have read and agreed to the published version of the manuscript.

**Funding:** This research was funded by the National Science Centre in Poland (project no. N N310 449538).

**Acknowledgments:** The author thanks reviewers for valuable comments that have enriched the content of the article.

**Conflicts of Interest:** The authors declare no conflict of interest. The funders had no role in the design of the study; in the collection, analyses, or interpretation of data; in the writing of the manuscript, or in the decision to publish the results.

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
