# Peer review of "Natural Compounds as Elicitors of Plant Resistance Against Diseases and New Biocontrol Strategies"

_agronomy, doi:10.3390/agronomy10020173_

Round 1

Reviewer 1 Report

Manuscript ID: Agronomy-654729

The author has written a nice review about a topic that has undergone tremendous progress in recent years. Many damage-associated molecular patterns (DAMPs), have been identified in different plant species that are able to trigger immune responses against pathogens with diverse life styles and also promote damage repair, minimizing losses caused by diseases (Hou et al., 2019). These findings have fostered an active area of investigation, because we are facing the challenge of fighting plant diseases, reducing the use of phytosanitary, and thus responding to the social demand to develop a more sustainable agriculture.

However, I have the following major points of criticism:

Concepts as PAMPs, MAMPs and DAMPs need to be previously well introduced in the text, as are the main sources of molecules able to elicit plant defence responses. Highlighting the potential commercial applications of the last ones (DAMPs), as those are plant derived molecules. In the same line, the term systemic acquired resistance (SAR) should be introduced. To my knowledge this type of resistance is triggered after the attack of an avirulent pathogen and the generation of a hypersensitive response (HR). In the text is not clear to me when the author is saying in lines 77-78 that “terpinen-4-ol and 1,8-cineole show synergistic antifungal activity”, if those assays have been done in vivo or in vitro and which are the pathogens that have been assayed. The same for the tea tree oil, has been used in fields or only in vitro?. Would be nice to know from the text which compounds have been analyzed only in vitro and which have been proved to be efficient in fields as biocontrol agents. Chitosan is a structural polysaccharide present in the shells of marine crustaceous, but also its important to mention that is the main component of fungal cell walls, and this PAMP is a strong elicitor of plant immune responses. Therefore, table 1 should include that information. Also, I would delete the chitosan as Line 99. Oligogalacturonides (OGAs), are small fragments of pectins that are released by the action of pathogen degrading enzymes, not by the action of bioactive compounds. β-1,3-glucan laminarin has been also identified as a molecular component of some necrotrophic fungus. Again, this PAMP has been demonstrated that works as a potent elicitor of plant defences (Mélida et al., The Plant Journal, 2017). Should be included in the table. The information about chitin perception should be fully updated. The molecular mechanisms underlying chitin perception have been described in the model plant Arabidopsis (Cao et al., 2014; Liu et al., 2012) and in rice (Hayafune et al., 2014; Kaku et al., 2006). The receptors have been identified as well as some of the downstream components activated after chitin or chitosan perception. Therefore, I would delete the sentences 87 to 93. Chitosan is obtained after deacetylation of chitin to increase its solubility. Waxes, lignin and suberin are not cellulose substances (Lines 136-137). This should be changed. Line 132: The sentence “Grapefruit extract used as a spray… “ should be moved to a new paragraph as it has no relation at all with the chitosan. Maybe should start a new paragraph focused on the relevant role of plant cell wall, and the new molecules identified able to activate plant defence responses, such as fragments of cutin (Buxdorf et al., 2014), pectins (OGAs; De Lorenzo et al., 2018), cellulose (Jonson et al., 2018) and hemicellulose (Claverie et al., 2018). This elicitor activity has been proved in Solanum lycopersicum, Glicine max, Nicotiana tabacum and Vitis vinifera. Would be nice to group the compounds that are commercialized, based on their active compound Eg chitin based (x), Algae extracts (x and y)...

I also have this minors:

Lines 41-42. When author says that natural bioactive compounds directly affect plant growth, seems that its application is deleterious for the plant. I would suggest to change to “They modulate or regulate plant growth”.

Line 104: change “grapefruit and algae extracts” instead of “grapefruit and algae“

Line 112-113: “The accumulation of phenolic compounds in a plant is the result of qualitative changes in the respiration process”. A reference should be included at the end of this sentence

Line 120: change “β-1,3-glucan laminarin” instead of “-1,3-glucan laminarin”

Line 138: I don’t understand what is a foreign factor.

Figure 1. I would suggest “Cell wall remodeling” instead of “epidermis thickening”

“Transcriptional reprogramming”” instead of “several gene expressions”. And from this box I would draw two arrows that pointed to the responses below BIOCHEMICAL RESPONSES and MORPHOLOGICAL AND ANATOMICAL RESPONSES

Line 156: A reference should be included at the end of this sentence

Line 159: change “recent research” instead of “modern research”

Line 169: not all of the natural elicitors have direct microbial action. Change it.

Line 209: 2006 should not be in italics

Author Response

A response to review 1

Chapter 2 The concepts of PAMP, MAMP, DAMP and the information about their importance as molecules eliciting defense responses, as well as definition of systemic acquired resistance (SAR) have been completed in the text, according to the reviewer's comments. Lines 77-78 with an unclear statement on the synergistic effect of terpinen-4-ol and 1,8-cineol were removed from the article due to the lack of more detailed information explaining this synergism. According to the reviewer's suggestions, the paper has been supplemented with information concerning the antimicrobial activity of the tested natural active compounds in laboratory and in vivo The text was supplemented with the information that chitosan is also a component of filamentous fungi cell walls, as well as a strong factor that elicits an immune response (MAMP). Line 99 has been removed according to the reviewer's comments.

Chapter 3

Mode of action of chitosan (updated information) was described in detail in Chapter 3. This was presented on the model example of Arabidopsis and rice, as suggested by the reviewer. Line 87-93 have been deleted, as suggested by the reviewer. Information in lines 136-137were corrected as suggested by the reviewer. Information concerning the role of the cell wall in plant immunological processes has been broadly supplemented according the literature proposed by the reviewer.

Chapter 4

Preparations based on natural components have been grouped in the text based on their active substance. Lines 41-42, 104, 112-113, 120, 138, 156, 159, 169, 209 have been corrected in accordance with the reviewer's comments. Figure 1 has been rebuilt in accordance with the reviewer's comments.

The article was corrected in accordance with the suggestions of the review. New literature items have been added to the article content, and references list have been changed. All changes are marked in red in the text.

Reviewer 2 Report

The manuscript reports “Natural compounds as elicitors of plant resistance against diseases and new biocontrol strategies.  The review work seems interesting to me although I think it lacks a bit more depth in studies of literature, need more citations to complete the review work and studies. There are several grammatical errors still exist in the text along with the flaws and ambiguity in several parts of the presentation. The manuscript  requires extensive revision.

1. Introduction  is too short. Keep more related citations.

2. Natural bioactive compounds as elicitors: Line 41-43- citations are missing.

3. Natural compounds against plant diseases: Line 60-63- citations are missing.

Table: You can give more example of natural bioactive compounds (origin and form) and their activity against pathogens.

Author Response

A response to review 2

The introduction has been supplemented with more related references, as suggested by the reviewer.

I do not agree with the reviewer's comment that the introduction is too short. In my opinion, this part of the work contains general information and purpose of the presented research. Therefore, it should not be developed too much, because they are simple and clear. Detailed and comprehensive information on the discussed topics (modes of action of natural active compounds), was provided in in Chapters 2-4. Therefore, the introduction has not been changed, but supplemented by additional citations.

2 and 3. Lines 41-43 and 60-63: missing quotation has been completed, as suggested by the reviewer.

The reviewer 2 suggested supplementing the table with other natural biocompounds.  The purpose of my report was to characterise only selected natural biologically active compounds and detailed describe their antimicrobial and immunological activity. Table completed with other examples of natural bioactive compounds, as suggested by the reviewer, would change significantly the structure of paper and its size, because in the text it should be describe mode of action of this all  compounds, as elicitors. It is difficult to consider all natural active compounds because they are very numerous.Therefore, the table has not been completed with other examples of bioactive compounds.

Round 2

Reviewer 1 Report

Manuscript ID: Agronomy-654729

The author has included all the changes suggested by the reviewer, and has done a good job integrating concepts and updating some recent findings. I have only found some little mistakes that I summarize below, but the manuscript is ready to be published once they are corrected.

Line 110, should be written chitin instead of chitosan, as chitin is the natural polymer

Line 112, on this sentence I would say that chitosan, that is more soluble than chitin, is obtained by the process that is described

Line 115, review grammar of the sentence

Line 118, activating their immune responses instead of causing their immune response

Line 129, extracts instead of extract

Line 182, references 57-58 instead of 56-56. The reference 57 should be Lorenzo not Lorenco

Line 182, delete than

Line 190, pathogens instead of pathogen

References need to be revised, as some journals are not in italics (line 334), some titles are in italics (line 342), some dots are missing (line 408).

Concerning figure 1, I really appreciate the changes, however I think I was not able to make me understand completely. I would leave the previous structure, that is, the plant generates 3 types of responses, i) biochemical, ii) morphological and iii) genetic. Not only one genetic response as in the second draft. But the transcriptional reprogramming affects the other two responses.

Author Response

Response to Reviewer 1 Comments

Point 1: Line 110, should be written chitin instead of chitosan, as chitin is the natural polimer; Line 112, on this sentence I would say that chitosan, that is more soluble than chitin, is obtained by the process that is described

Response 1: Lines 110 and 112 have been corrected in accordance with the reviewer's comments.

Point 2: Line 115, review grammar of the sentence

Response 2: The sentence has been corrected forField studies confirmed the effectiveness of chitosan micro-gel (0.1%) for inhibition of soybean diseases caused by soil-borne pathogenic fungi (Fusarium spp., Pythium spp., Botrytis cinerea, and Rhizoctonia solani).”

Point 3: Line 118, activating their immune responses instead of causing their immune response; Line 129, extracts instead of extract; Line 182, references 57-58 instead of 56-56. The reference 57 should be Lorenzo not Lorenco; Line 182, delete than; Line 190, pathogens instead of pathogen

Response 3: Lines 118, 129, 182, 182, 190 as well as references 57 have been corrected in accordance with the reviewer's comments. 

Point 4: References need to be revised, as some journals are not in italics (line 334), some titles are in italics (line 342), some dots are missing (line 408).

Response 4: All references was revided and completed according the Instruction for the authors. Line 342: no changes were made because  according the Instruction for the authors the book title should be in italics and year not in bold. Changes in references 10, 11, 27, 45, 53, 67, 77 have been made. All changes in the text are marked in red.

Point 5: Concerning figure 1, I really appreciate the changes, however I think I was not able to make me understand completely. I would leave the previous structure, that is, the plant generates 3 types of responses, i) biochemical, ii) morphological and iii) genetic. Not only one genetic response as in the second draft. But the transcriptional reprogramming affects the other two responses.

Response 5: Figure 1 has been rebuilt in accordance with the reviewer's comments and the figure 1 shows the influence of genetic modifications on changes in plant morphology and biochemistry.
